# Three Lipid Emulsions Reduce *Staphylococcus aureus*-Stimulated Phagocytosis in Mouse RAW264.7 Cells

**DOI:** 10.3390/microorganisms9122479

**Published:** 2021-11-30

**Authors:** Ming-Shan Chen, Yi-Wei Tung, Chia-Lin Hu, Hui-Ju Chang, Wen-Chun Lin, Shew-Meei Sheu

**Affiliations:** 1Department of Anesthetics, Ditmanson Medical Foundation, Chia-Yi Christian Hospital, Chia-Yi City 60002, Taiwan; 06590@cych.org.tw (M.-S.C.); 07442@cych.org.tw (Y.-W.T.); 07389@cych.org.tw (C.-L.H.); 2Department of Medical Laboratory Science and Biotechnology, Asia University, Taichung City 41354, Taiwan; 3Department of Medical Research, Ditmanson Medical Foundation, Chia-Yi Christian Hospital, Chia-Yi City 60002, Taiwan; 14737@cych.org.tw (H.-J.C.); 13320@cych.org.tw (W.-C.L.)

**Keywords:** *S. aureus*, phagocytosis, Lipofundin, macrophage, ROS

## Abstract

Soybean oil (SO)-, SO medium-chain triglyceride (MCT)-, olive oil (OO)-, and fish oil (FO)-based lipid emulsions are generally applied in clinical practice via intravenous injection for patients with nutritional requirements. The function of lipid emulsions on immune modulation remains inconsistent, and their effects on macrophages are limited. In the present study, we used a model of *S. aureus*-infected mouse RAW264.7 macrophages to determine the influence of three different compositions of lipid emulsions (Lipofundin, ClinOleic, and Omegaven) on reactive oxygen species (ROS) production, phagocytosis, and bacterial survival. The three individual lipid emulsions similarly enhanced bacterial survival but reduced *S. aureus*-stimulated ROS, phagocytosis of *S. aureus* bioparticles conjugate, polymerization of F-actin, and phosphorylation of AKT, JNK, and ERK. Compared with the JNK and ERK inhibitors, the PI3K inhibitor markedly suppressed the phagocytosis of *S. aureus* bioparticles conjugate and the polymerization of F-actin, whereas it significantly increased the bacterial survival. These results suggest that the three lipid emulsions diminished ROS production and phagocytosis, resulting in increased bacterial survival. PI3K predominantly mediated the inhibitory effects of the lipid emulsions on the phagocytosis of mouse RAW264.7 macrophages.

## 1. Introduction

Lipid emulsions for clinical use are usually injected intravenously for nutrition for critically ill patients [1]. Soybean oil (SO)-, SO-medium chain triglyceride (MCT)-, olive oil (OO)-, and fish oil (FO)-based lipid emulsions are generally recommended in clinical practice for the lipid supply. The immune modulation activity of lipid emulsions used in clinical applications is still controversial [2,3,4].

Different compositions of fatty acids in lipids influence several features of immune cells, including the cell membrane structure, production of bioactive substances, intracellular signaling pathways, regulation of gene expression, and immune modulation [2,4]. Phagocytes are innate immune cells that are involved in protection against infection [5,6]. One of their major functions, phagocytosis, is an important mechanism to remove pathogens and cell debris. The pathogen is trapped in a phagosome, which fuses with a lysosome to form a phagolysosome. Within the phagolysosome, enzymes and reactive oxygen species (ROS) digest the pathogen [6].

*Staphylococcus aureus* (*S. aureus*) is an important pathogen that causes human morbidity and mortality throughout the world [7]. It leads to various infections ranging from minor skin infections and soft tissue infections to life-threatening invasive diseases, such as necrotizing pneumonia and sepsis [7,8,9]. *S. aureus* has several mechanisms to escape host immune defenses and establish an infection, including resistance to antimicrobial peptides, clearances of ROS, inhibition of complement activation and neutrophil recruitment, and evasion of phagocytosis [10,11].

Macrophage responses to the intravenous infusion of lipid emulsions are diverse due to the various experimental designs and the administered doses in animal and human studies [12]. An important function of macrophages is to control a bacterial infection. The origins of macrophages and whether macrophages are activated or not are also attributed to the modulation effect of lipid emulsions [12,13,14]. In this study, we compare the immunomodulation effects of different lipid emulsions on the same model of bacteria-infected macrophages. The effects of three parenteral emulsions containing different lipid compositions (Lipofundin, ClinOleic, and Omegaven) on ROS production, phagocytosis, and bacterial survival in *S. aureus*-infected RAW264.7 macrophages are investigated.

## 2. Materials and Methods

### 2.1. Bacteria, Cell Culture, and Bacterial Infection 

*S. aureus* ATCC 25923 cultured on tryptic soy agar with 5% sheep blood was refreshed in Difco^TM^ Luria–Bertani (LB) broth (Becton, Dickinson and Company, Sparks, MD, USA) for 16 h to further perform the infection assay. The RAW264.7 mouse macrophage cell line was purchased from the Food Industry Research and Development Institute in Taiwan. The RAW264.7 cells were grown in DMEM (Gibco-BRL, Grand Island, NY, USA) supplemented with 10% fetal calf serum (FCS, Gibco-BRL) and sub-cultured every second to third day. According to the established infection method [15], a 10-fold dilution of *S. aureus* suspension (OD_600_ = 1) was incubated with RAW264.7 cells for 30 min. Then, the free bacteria were washed away with 1× phosphate-buffered saline (PBS) to set up an infected condition with a multiplicity of infection of 15. The cells were incubated in a medium with 2% FCS for further analysis. 

### 2.2. Reagents

Lipofundin 20% is composed of soybean oil (100 g/L), medium-chain triglycerides (MCT, 100 g/L), egg lecithin, glycerol, α-tocopherol, and sodium oleate. Lipofundin 20% was purchased from B. Braun Melsungen AG (Melsungen, Germany). ClinOleic 20% was obtained from Baxter (Norfolk, UK), which contains a mix of refined olive oil and refined soybean oil (200 g/L), glycerol, egg phospholipids, sodium oleate, and sodium hydroxide. Omegaven (Fresenius Kabi Austria GmbH, Graz, Austria) is a pure fish oil emulsion supplement (100 g/L) containing a high percentage of eicosapentaenoic acid (EPA) and docosahexaenoic (DHA). Di-α-tocopherol, glycerol, egg phosphatides, sodium oleate, and sodium hydroxide are also included in it. The amount of Lipofundin (60 μg/mL) was chosen because the 10-fold concentration was applied to dissolve the clinical relevant concentration of anesthetics (propofol: 6 μg/mL). The same concentrations of ClinOleic and Omegaven were used to conduct the following experiments. The NADPH oxidase inhibitor, diphenyleneiodonium chloride (DPI) was obtained from Abcam (ab141310, Cambridge, UK) and used to inhibit ROS production. PD98059, a specific inhibitor of mitogen-activated protein kinase, was purchased from TargetMol (Boston, MA, USA). LY294002 is a specific cell-permeable phosphatidylinositol 3-kinase (PI3K) inhibitor (Sigma-Aldrich, Inc., St. Louis, MO, USA). SP600125 is a selective inhibitor of c-Jun N-terminal kinase (JNK) (Sigma-Aldrich).

### 2.3. Luminol Chemiluminescence Assay

Luminol (5-amino-2,3-dihydro-1,4-phthalazindione) was applied to measure the total amount of intra- and extracellular ROS of the RAW264.7 cells (1 × 10^5^ cells/well) in a 96-well white microplate. After preincubation with lipid emulsions (60 μg/mL) or DPI (10 μM) for 30 min, *S. aureus* was suspended with the same concentrations of lipid emulsions or DPI, which were added to the cells to stimulate ROS production for 30 min. After being washed twice with 1× PBS, the cells were immediately stained with 0.05 mg/mL luminol (Sigma-Aldrich). The response of the chemiluminescence resulting from ROS production was continuously measured over 1 h at 37 °C by using a BioTek Synergy H1 microplate reader (BioTek, Winooski, VT, USA). *S. aureus*-infected RAW264.7 cells without the lipid emulsion pretreatment were used as a positive control. The basal controls were the cells treated with each lipid emulsion alone. The relative effect of the lipid emulsions on *S. aureus*-stimulated ROS production was calculated by the peak value of chemiluminescence at 5 min.

### 2.4. Determination of Intracellular HOCl

BioTracker™ TP-HOCL 1 dye (EMD Millipore Corporation, Temecula, CA, USA) is a highly selective and sensitive fluorescent probe for imaging HOCl in live cells. RAW264.7 cells (5 × 10^5^ cells/well) suspended in DMEM supplemented with 10% FCS were plated in a Nunc^TM^ 177437 Lab-Tek Chamber Slide System. After overnight incubation, the cells were washed with 1×PBS once and preincubated with or without lipid emulsions for 30 min. Then, *S. aureus* suspended with or without lipid emulsions in DMEM (2% FCS) containing BioTracker TP-HOCL 1 live cell dye (20 μM) was added to the cells and maintained at 37 °C for 30 min. The blank cell control presented the basal HOCl production. Fluorescent images were taken with a 10× objective by an LSM800 confocal microscope (Carl Zeiss Microscopy GmbH, Jena, Germany), and the intensity of the HOCl fluorescence was obtained from the calculation of the images. 

### 2.5. Phagocytosis Assay

Phagocytosis was quantified by the pHrodo™ Green *S. aureus* BioParticles^®^ conjugate (P35367, Life Technologies, Carlsbad, CA, USA) based on the pHrodo™ dye conjugated bioparticles, which dramatically increase green fluorescence at an acidic pH. First, the RAW264.7 cells were suspended in DMEM medium with 10% FCS and loaded in a 96-well plate (1 × 10^5^ cells/well) for incubation overnight. After pretreating with lipid emulsions for 30 min, pHrodo Green *S. aureus* bioparticles resuspended in a live cell imaging solution (100 μg/mL) with or without lipid emulsions were applied to the cells. Cells that did not undergo lipid pretreatment and were only incubated with *S. aureus* bioparticles were used as the positive control. The plates were covered and maintained at 37 °C for 3 h in the absence of CO_2_. The fluorescence was recorded at 490-nm excitation and 540-nm emission wavelengths using a BioTek Synergy H1 microplate reader (BioTek). In order to visualize the acidified compartments, the cells that had been incubated with *S. aureus* bioparticles were further stained with Hoechst 33342 (2 μg/mL) and 100 nM LysoTracker Red DND-99 (Invitrogen Life Technologies, Carlsbad, CA, USA). Phagocytosis in the live cell imaging was visually determined with a fluorescence microscope (Olympus Corporation, Tokyo, Japan). Phagocytosis images of the cells treated with inhibitors were obtained from an LSM800 confocal microscope (Carl Zeiss Microscopy GmbH).

### 2.6. Confocal Immunofluorescence Microscopy

For the investigation of F-actin, cells (1 × 10^5^) were grown on a dish with a glass bottom overnight. After pretreating with or without lipid emulsions for 30 min, *S. aureus* combined with or without lipid emulsions were subsequently added to the cells for a further 30 min of incubation. The supernatant was discarded, while the remaining cells were washed twice with 1× PBS and fixed with 4% formaldehyde for 10 min at room temperature. Cells were permeabilized with 0.1% Triton X-100 in PBS for 5 min and then incubated in Image-iT™ FX Signal Enhancer (I36933) for 30 min. The actin was selectively labeled with phalloidin–Alexa Fluor 568 (Life Technologies, Eugene, OR, USA) for 20 min. Phalloidin is a high-affinity F-actin binding probe which is conjugated to the photostable dye Alexa Fluor 568. Afterward, the cells were stained with 300 nM of DAPI (Sigma-Aldrich), mounted with fluorescent mounting medium (Fluoroshield^™^ with 1,4-Diazabicyclo[2.2.2]octane, Sigma-Aldrich), and subsequently visualized under the LSM800 confocal microscope (Carl Zeiss Microscopy GmbH) to obtain the images. Before being stained with each solution, the cells were washed twice with 1× PBS. The structure of the filopodial membrane protrusion could be identified through F-actin staining with phalloidin–Alexa Fluor 568.

### 2.7. S. aureus-Infected RAW264.7 Cells with the Treatment of Lipid Emulsions for Western Blot Analysis

RAW264.7 cells (1.5 × 10^6^ cells/well) were seeded in 6-well plates and left overnight. After washing with 1× PBS, lipid emulsions dissolved in DMEM medium with 2% FCS were added to RAW264.7 cells for 30 min, and then *S. aureus*, suspended in the medium with the same concentration of lipid emulsions (1.5 mL), was applied to the cells for a further 60 min. After incubation, the cells were washed with 1× PBS and collected for western blot analysis. Mammalian protein extraction reagent (Thermo Scientific Inc., Rockford, IL, USA) containing a 0.1% protease inhibitor cocktail was used to lyse the cell pellets, where equal proteins (40 μg) in each sample were subjected to SDS-PAGE gels and electrotransferred to the PVDF membranes. The primary antibodies associated with the phagocytic signal pathways were selective for AKT (#4691), phospho-AKT (Ser 473) (#4060), JNK (#9258), p-JNK (Thr183/Tyr185) (#4668), p44/42 MAPK (ERK1/2) (#4695), and phospho-p44/42 MAPK (Thr202/Tyr204). Anti-GAPDH (GeneTex, Inc., Irvine, CA, USA) antibody was used as an internal control. After incubation with secondary antibodies conjugated to horseradish peroxidase (HRP), the blot was developed with Immobilon Western Chemiluminescent HRP substrate (EMD Millipore Corporation, Billerica, MA, USA) and detected using a BioSpectrum Imaging System (UVP).

### 2.8. Bacterial Survival Assay

Bacterial survival assays were conducted according to the method described in the published paper [15]. RAW264.7 cells that had undergone pretreatment with lipid emulsions were infected with *S. aureus* at the same concentration of the lipid emulsions for a further 30 min. Cells infected with *S. aureus* but without lipid emulsion treatment were used as a reference control. The cell-associated bacteria at the start of infection (0 h) and viable bacteria at the end of infection (3 h) were counted by serial dilution and grown on LB agar plates.

## 3. Results

### 3.1. The Effect of Lipid Emulsion Treatment on S. aureus Survival

To evaluate *S. aureus* survival under lipid emulsion treatment (60 μg/mL), we counted the cell-associated bacterial number at the start and end of infection. At the starting time point, the bacterial numbers were similar, even for the *S. aureus*-infected RAW264.7 cells treated with different lipid emulsions (Figure 1a). At the end of infection, the bacterial number of infected RAW264.7 cells was significantly increased in those that received the three lipid emulsion treatments compared with the untreated *S. aureus*-infected cells (Figure 1b).

### 3.2. Lipid Emulsions Decreased the ROS Levels

The generation of ROS is a major antimicrobial activity of macrophages. The time kinetics of the *S. aureus*-stimulated total ROS is shown in Figure 2a, and the peak value occurred at 5 min. All three lipid emulsions significantly diminished the total ROS production of *S. aureus*-infected RAW264.7 cells (Figure 2a), and the luminescence value related to the positive control was further calculated, as shown in Figure 2b. Treatment with either Lipofundin or ClinOleic (60 μg/mL) reduced approximately 60% of the ROS production. Omegaven minimized more ROS production than Lipofundin or ClinOleic and reached a level significantly lower than that of Lipofundin. Specific intracellular HOCl, which expresses potent antimicrobicidal activity against the phagocytized bacteria, was stained with the fluorescence probe (BioTracker TP-HOCL 1 live cell dye). The *S. aureus*-infected RAW264.7 cells displayed strong fluorescence, whereas the fluorescence enhancement could be largely inhibited by the three lipid emulsions (Figure 2c).

### 3.3. Lipid Emulsions Decreased the Phagocytosis of S. aureus Bioparticles

To further examine whether the lipid emulsions increased bacterial survival by decreasing phagocytosis, green *S. aureus* bioparticles were used to detect phagocytosis. We found that the three lipid emulsions (60 μg/mL) significantly inhibited the phagocytosis of green *S. aureus* bioparticles. Moreover, their suppression (>40%) was similar (Figure 3a), with the phagocytosis shown in Figure 3b. The green *S. aureus* bioparticles that colocalized with lysosome (red) was decreased in RAW264.7 cells with lipid emulsion treatment. The intensity of LysoTracker in RAW264.7 cells treated with lipid emulsions was comparable to that without treatment, indicating that the acidified compartments were intact even after the lipid emulsion treatments (Figure 3b).

Phagocytosis was driven by the reorganization of filamentous actin (F-actin), which was visualized by staining with fluorescently labeled phalloidin. The major morphology of the RAW264.7 cells in the blank control possessed a clearly rounded F-actin ring (Figure 4). *S. aureus* infection stimulated the production of numerous membrane protrusions (actin-rich filopodia) which caused the macrophages to increase in cell area. Treatment with cytochalasin D, a drug that inhibits the reorganization of actin filaments, decreased the membrane protrusions and actin-rich filopodia. The cells treated with lipid emulsions actually suppressed the *S. aureus*-infected morphology of F-actin in a way close to the cytochalasin D treatment.

### 3.4. PI3K/AKT and JNK Are Involved in the Lipid Emulsion-Mediated Inhibition of Phagocytosis

JNK activation is involved in *S. aureus*-infected phagocytosis by RAW264.7 cells [16]. *S. aureus* co-incubated RAW264.7 actually induced a higher phosphorylation level of JNK in a time-dependent manner over 30–90 min (Figure 5a). The time point (60 min) of *S. aureus* infection was applied to further experiments. We also found that Lipofundin could inhibit the phosphorylation level of JNK (Figure 5a). To investigate whether lipid emulsions could inhibit different kinases except JNK, the RAW264.7 cells were treated with lipid emulsions prior to and during *S. aureus* infection. The phosphorylated AKT, JNK, and ERK were obviously inhibited by three lipid emulsions (Figure 5b). We further used the inhibitors of PI3K (LY294002), JNK (SP600125), and ERK (PD98059) to block their activation and then determined their effect on the phagocytosis of *S. aureus*. To compare the phagocytosis in the absence and presence of inhibitors, the inhibition of PI3K showed an almost 60% reduction in the phagocytosis of *S. aureus* bioparticles by the RAW264.7 cells (Figure 5c,d). LY294002 treatment also diminished *S. aureus*-stimulated membrane protrusions and actin-rich filopodia in a similar manner to cytochalasin D treatment (Figure 5e). An inhibitor of JNK (SP600125) reduced around 20% of the phagocytosis of *S. aureus*, but an inhibitor of ERK (PD98059) only slightly decreased the phagocytosis by RAW264.7 cells (Figure 5c,d). Finally, bacterial survival was significantly increased when LY294002 was incubated with *S. aureus*-infected RAW264.7 cells. SP600125 slightly promoted bacterial survival (Figure 5f). These data implied that PI3K plays a major role in the lipid emulsion-mediated inhibition of phagocytosis to promote bacterial survival.

## 4. Discussion

SO-, SO-MCT-, OO-, and FO-based lipid emulsions are generally used for critically ill patients for the supply of lipid nutrition. However, the immune modulation activity of lipid emulsions is still controversial [2,3,4]. In the present study, we used the model of *S. aureus*-infected mouse RAW264.7 macrophages to investigate the effects of three lipid emulsions (Lipofundin, ClinOleic, and Omegaven) on the antimicrobial activity of macrophages. The three lipid emulsions similarly increased the level of bacterial survival; even Omegaven caused lower ROS production. All lipid emulsions also reduced the phagocytosis of *S. aureus* bioparticles conjugate, the polymerization of F-actin, and the expression of p-AKT, p-JNK, and p-ERK. We further demonstrated that the PI3K inhibitor obviously suppressed the phagocytosis of *S. aureus* bioparticles conjugate and the polymerization of F-actin, which were similar to the effect of cytochalasin D, a polymerization inhibitor of actin used to block phagocytosis. These results indicate that the three lipid emulsions diminished ROS production and phagocytosis, leading to the increase in bacterial survival in mouse RAW264.7 macrophages. PI3K is a key regulator in the lipid emulsion-mediated inhibition of phagocytosis in such an infection model.

Lipofundin, ClinOleic, and Omegaven are SO-MCT-, OO-, and FO-based lipid emulsions. We first found that all three lipid emulsions increased *S. aureus* survival in a RAW264.7 infection (Figure 1), in agreement with Miles et al. [3]. They suggested that there is little difference in immune modulation between SO, SO-MCT, and OO. The ratio of *n*-6 to *n*-3 polyunsaturated fatty acids (PUFAs) of Omegaven (1:8) is much lower than that of Lipofundin (7:1) and ClinOleic (9:1). Our data challenged the view that the immune modulation by lipids is determined by the adjustment of the ratio of *n*-6 to *n*-3 PUFAs [2]. Grimble et al. suggested that the highest production of lipid mediators is dependent on the optimal ratio of *n*-6 to *n*-3 PUFAs rather than a linear correlation [17].

Lipofundin (2.5 mM-3 mM, SO-MCT-based lipid emulsion) has been proven to enhance ROS production in PMA- or FMLP-stimulated human neutrophils [18,19]. However, our previous study demonstrated that a clinically relevant concentration (60 μg/mL, 94.64 μM) of Lipofundin can decrease the total ROS of *S. aureus*-infected RAW264.7 cells [20]. In the present study, three different lipid emulsions were assessed in one system to compare their effects on *S. aureus*-infected mouse macrophages. The same concentration (60 μg/mL) of ClinOleic and Omegaven (OO- and FO-based lipid emulsions) was also consistent in the suppression of total ROS production. In particular, Omegaven caused a significant ROS reduction compared with Lipofundin (Figure 2b). A similar concentration of Nutrilipid (SO-based, 100% LCT) was found to diminish the ROS production of PMA-activated human neutrophils [21]. Taken together, the concentration of lipid emulsion influences ROS production by activated phagocytes more than the composition of lipid emulsion.

Phagocytosis is a defense mechanism used by macrophages to clear pathogens, and it is strictly dependent on actin polymerization. Lipofundin has the ability to inhibit phagocytosis [20]. Figure 3 additionally shows the suppression of ClinOleic and Omegaven on the phagocytosis of *S. aureus* and the attenuation of F-actin polymerization (Figure 4). The JNK signal pathway mediates the *S. aureus*-stimulated phagocytosis of RAW264.7 cells [16]. We confirm that *S. aureus* infection induced a time-dependent activation of JNK (Figure 5a) in mouse RAW264.7 cells, and the inhibition of JNK reduced the phagocytosis of *S. aureus* by around 20% (Figure 5c,d). To further investigate the kinases involved in the lipid emulsion-mediated inhibition of phagocytosis, we found that the phosphorylation level of AKT, JNK, and ERK suppressed by all three lipid emulsions was close to that of the inhibitor of phagocytosis: cytochalasin D (Figure 5b).

Treatment with a specific inhibitor of PI3K (upstream kinase of AKT) led to more inhibition of *S. aureus*-stimulated phagocytosis by RAW264.7 cells than that with JNK or ERK inhibitors (Figure 5c,d). Fang et al. demonstrated that the PI3K inhibitor reduced the partial phagocytosis of *S. aureus* by RAW264.7 cells [22], whereas the inhibition of JNK activation reduced the phagocytosis of *S. aureus* by 60% in their other study [16]. The authors did not compare the phagocytosis of *S. aureus* by RAW264.7 cells with the treatments of JNK or PI3K inhibitors at the same time. In addition to comparing the inhibitions of JNK and PI3K in our phagocytosis assays (Figure 5c,d), inhibition of PI3K activity also obviously diminished F-actin polymerization and significantly raised the bacterial survival (Figure 5e,f). These results suggest that PI3K plays a more dominant role than JNK in *S. aureus*-stimulated phagocytosis by mouse RAW264.7 macrophages, which is further related to the increase in bacterial survival.

## 5. Conclusions

All three lipid emulsions enhanced bacterial survival in the mouse RAW264.7 macrophages by a similar level, which may have been caused by their inhibition of *S. aureus*-activated ROS production, polymerization of F-actin, and phagocytosis. PI3K was dominantly involved in the suppression pathway of phagocytosis and bacterial survival (Figure 6). The concentration of lipid emulsions is more critical than the composition of lipid emulsions in our phagocytosis model.

## Figures and Tables

**Figure 1 microorganisms-09-02479-f001:**
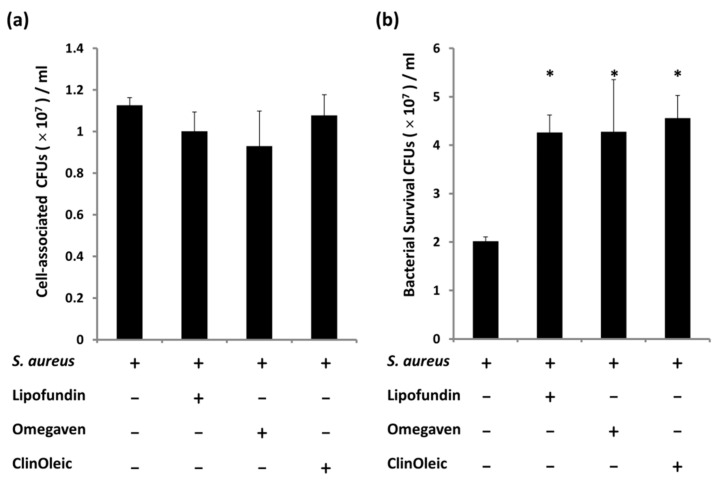
Treatment with lipid emulsions increased *S. aureus* survival in RAW264.7 cells. (**a**) The cell-associated colony-forming units (CFUs) at the start of infection (0 h). (**b**) Bacterial survival at the end of infection (3 h). The three lipid emulsions (60 µg/mL) were used to treat cells prior to and during *S. aureus* infection. After bacterial adherence for 30 min, the cells were washed with 1× PBS, and the cell-associated CFUs were counted at this time. The rest of the cells were incubated with a medium supplemented with lipid emulsions, and the bacterial survival CFUs were counted after 3 h. The data are shown as the means ± SD of three individual experiments. Significance was analyzed by one-way ANOVA followed by Bonferroni test (*p* < 0.05). * A significant difference compared with the *S. aureus*-infected control.

**Figure 2 microorganisms-09-02479-f002:**
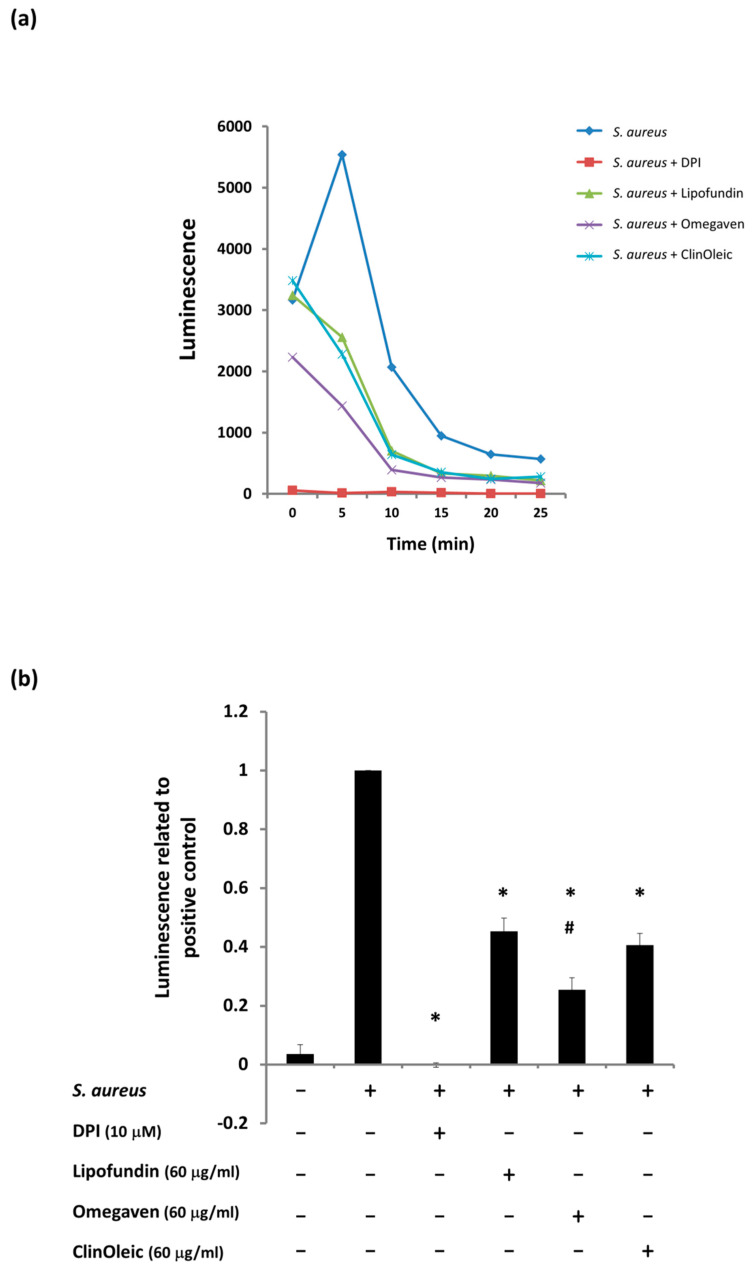
Three lipid emulsions inhibited ROS production and intracellular HOCl. Mouse RAW264.7 cells were treated with or without lipid emulsions prior to and during *S. aureus* infection. (**a**) The time kinetics of the total ROS in *S. aureus*-infected RAW264.7 cells were detected by luminol chemiluminescence assays. (**b**) The relative fold changes were the net luminescence value induced by co-incubation with the indicated agents and bacteria divided by the luminescence stimulated by *S. aureus*. The means ± SD of the three experiments are given. * = A significant difference compared with the *S. aureus*-infected control. # = A significant difference between the treatments with Lipofundin and Omegaven. (**c**) Intracellular HOCl stained by BioTracker TP-HOCL 1 live cell dye. Diphenyleneiodonium chloride (DPI, 10 μM) was used to block ROS production as a negative control. Fluorescent images were taken by confocal microscopy with a 10× objective. The scale bar is 100 µm.

**Figure 3 microorganisms-09-02479-f003:**
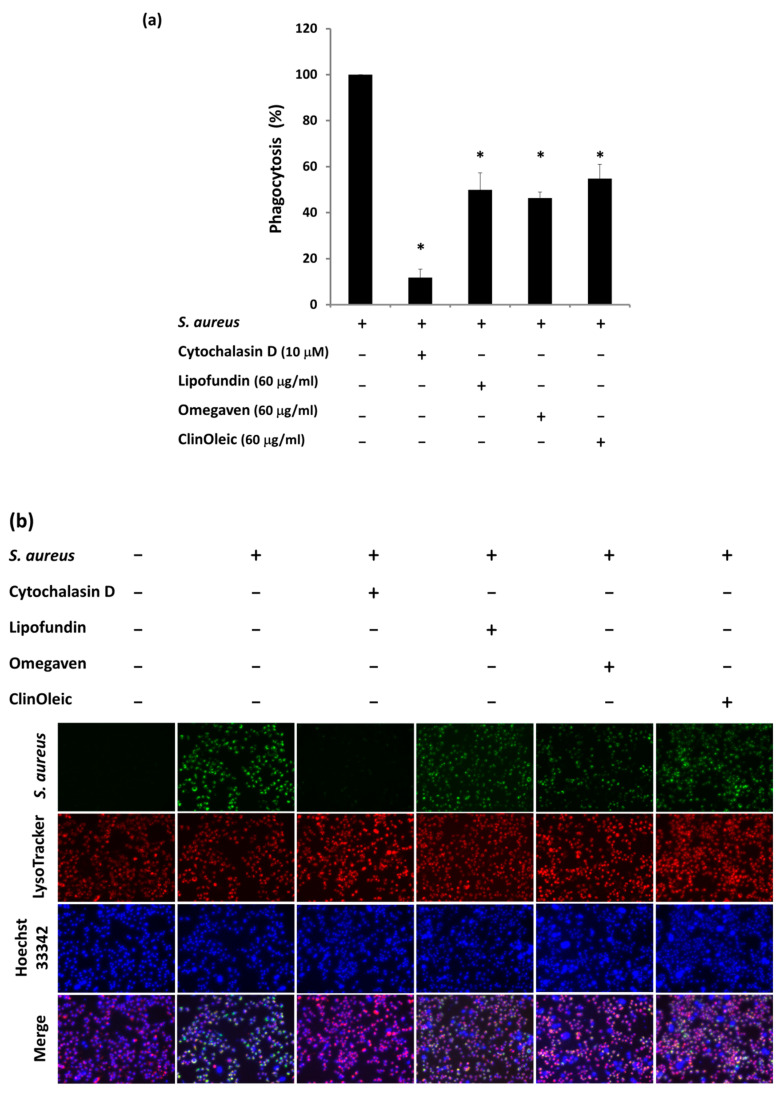
Three lipid emulsions repressed the phagocytosis. (**a**) Applying pHrodo™ Green *S. aureus* Bioparticles^®^ conjugate in the absence or presence of indicated reagents to quantify the phagocytosis of mouse RAW264.7 macrophages. Net phagocytosis was calculated by subtracting the fluorescence intensity of the control wells without cells from that of *S. aureus* bioparticles with or without lipid emulsions and the cytochalasin D (phagocytosis inhibitor, 10 µM) treatment wells. The phagocytic ability related to the *S. aureus* bioparticles was determined. The data are presented as the means ± SD of three individual experiments. * Significant differences compared with the positive control by one-way ANOVA followed by Bonferroni test (*p* < 0.05). (**b**) Live cell imaging of phagocytosis. The nuclei of RAW264.7 cells that were incubated with green *S. aureus* bioparticles were stained with Hoechst 33342 (blue). The lysozymes were labeled with LysoTracker (red). The images were obtained by fluorescence microscopy with a 10× objective.

**Figure 4 microorganisms-09-02479-f004:**
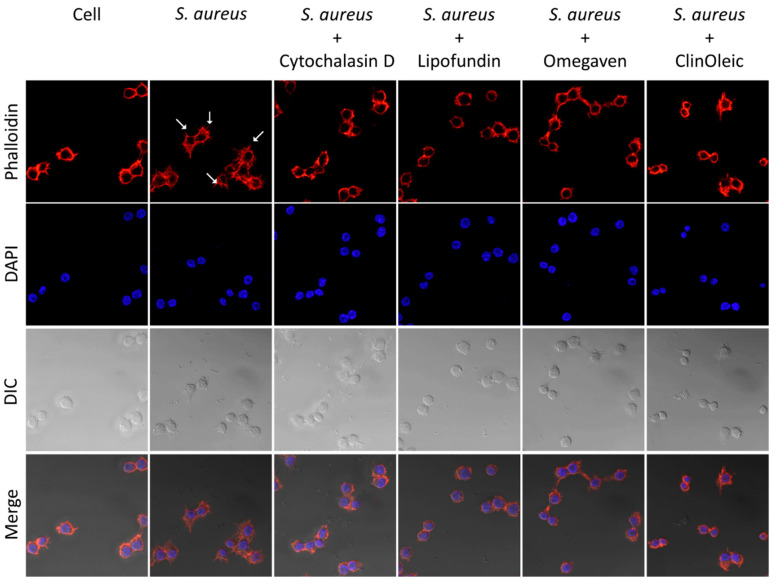
The three lipid emulsion-induced morphology changes in the cytoskeleton. *S. aureus*-infected mouse RAW264.7 macrophages were treated with or without lipid emulsions and further stained with Alexa-Fluor 568-labeled phalloidin to visualize F-actin. Cytochalasin D (10 µM) was used to block the phagocytosis of *S. aureus*. Cell morphology was observed by confocal microscopy (63×). The white arrow is filopodia.

**Figure 5 microorganisms-09-02479-f005:**
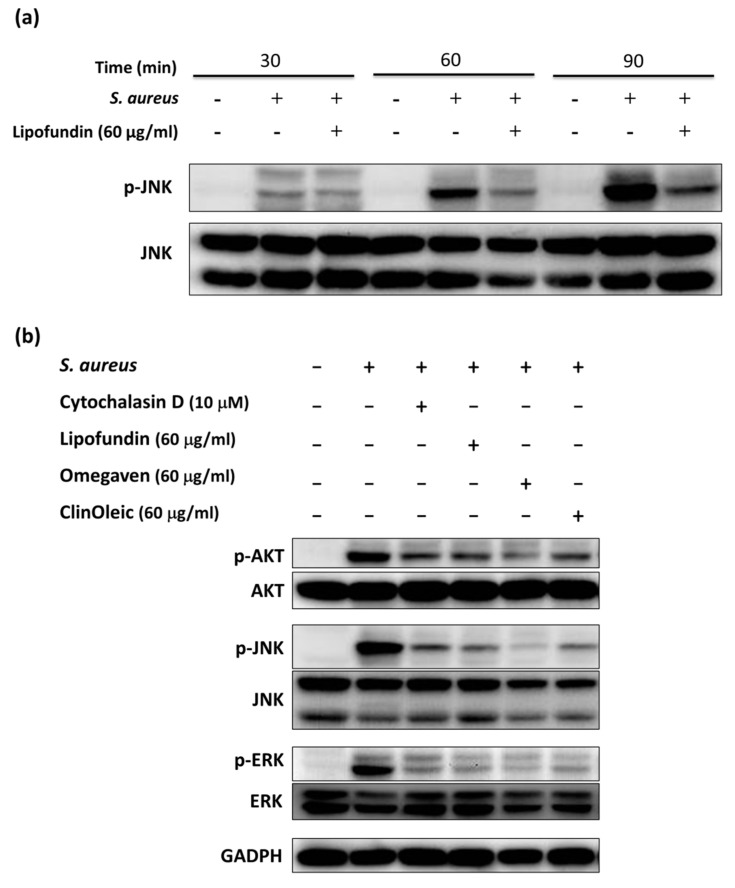
The activation of PI3K and JNK is involved in the lipid emulsion-caused reduction in phagocytosis. (**a**) The increase of p-JNK in *S. aureus*-infected RAW264.7 cells during the indicated time points was detected by western bolt analysis. (**b**) The phosphorylation of AKT, JNK, and ERK in *S. aureus*-infected RAW264.7 cells with or without lipid emulsion treatment was determined by western blot analysis. (**c**) The inhibitors’ effects on the phagocytosis of RAW264.7 cells were quantified by pHrodo™ Green *S. aureus* Bioparticles^®^ conjugate in the absence (positive control) or presence of indicated reagents. The reagents included the inhibitors of PI3K (LY294002), JNK (SP600125), ERK (PD98059), and phagocytosis (cytochalasin D). (**d**) Image of phagocytosis in live cells. RAW264.7 cells were treated with and without inhibitors prior to and during phagocytosis. The nuclei of cells were stained with Hoechst 33342 (blue). The location of the lysozymes was indicated by LysoTracker (red). The images were obtained by confocal microscopy with a 10× objective. (**e**) The inhibitors induced the changes in F-actin polymerization. Alexa-Fluor 568-labeled phalloidin was used to visualize the F-actin of *S. aureus*-infected RAW264.7 cells in the absence or presence of inhibitors, which was observed by confocal microscopy (63×). White arrow = filopodia. (**f**) Treatment with PI3K inhibitors significantly increased *S. aureus* survival. *S. aureus*-infected RAW264.7 cells were treated with or without the indicated reagent. At the start of infection (0 h), the cell-associated colony-forming units (CFUs) were similar between different treatments. Bacterial survival CFUs were grown and counted at the end of infection (3 h). The data are shown as the means ± SD of three individual experiments. Significance was analyzed by one-way ANOVA followed by the Bonferroni test (*p* < 0.05). * Significant difference compared with the *S. aureus*-infected control.

**Figure 6 microorganisms-09-02479-f006:**
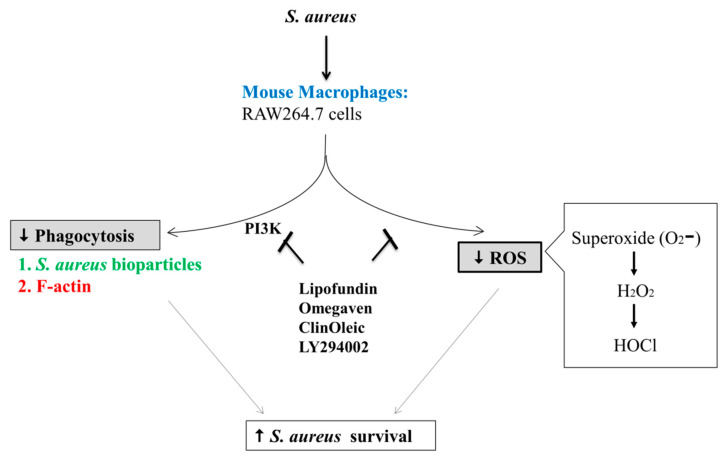
The in vitro inhibition model of lipid emulsions on *S. aureus*-infected mouse RAW264.7 macrophages. The three lipid emulsions increased *S. aureus* survival through the suppression of ROS and phagocytosis. Treatment with PI3K inhibitor (LY294002) reduced F-actin reorganization and phagocytosis, which supports the three lipid emulsion-mediated inhibition mechanisms.

## Data Availability

The generated and analyzed data in the present study supporting the findings are included within the article.

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
