# Peer review of "Three Lipid Emulsions Reduce Staphylococcus aureus-Stimulated Phagocytosis in Mouse RAW264.7 Cells"

_microorganisms, 2021, doi:10.3390/microorganisms9122479_

Round 1

Reviewer 1 Report

In this study the authors aim to investigate three parenteral emulsions containing different lipid compositions (Lipofundin, Clinoleic and Omegaven) sepa-54 rately on ROS production; phagocytosis and bacterial survival in S. aureus infected 55 RAW264.7 macrophages. The work is well prepared, providing a large amount of data and results.

I have only some minor concerns:
From what I have read, I do not understand how the effects of the emulsions were tested. I suggest adding a graphic scheme in the body of manuscript or as a Supplementary Materials.

Author Response

Reviewer 1:

In this study the authors aim to investigate three parenteral emulsions containing different lipid compositions (Lipofundin, Clinoleic and Omegaven) sepa-54 rately on ROS production; phagocytosis and bacterial survival in S. aureus infected 55 RAW264.7 macrophages. The work is well prepared, providing a large amount of data and results.

I have only some minor concerns:

From what I have read, I do not understand how the effects of the emulsions were tested. I suggest adding a graphic scheme in the body of manuscript or as a Supplementary Materials.

Ans: Thank you for the suggestion. We have added Figure 6 to describe the effects of three lipid emulsions that we evaluated.

Reviewer 2 Report

Chen et al. manuscript discusses the potential effects of the use of lipid emulsions in clinical practice for IV injection in patients requiring IV nutrition through in vitro study on macrophage cell line (RAW264.7) infected with Staphylococcus aureus.

Starting from the abstract, it was difficult to read this manuscript especially the results that were unclear mainly due to the language and style. Thus, it was difficult to evaluate the manuscript with the use of such obscure language. Extensive editing of English language and style is required.

The introduction does not provide sufficient background and needs major revision to its content as well as language.

The methods must be improved and details must be described for appriopriate duplication of methods and hence results.

There was no mention of how the authors came up with a MOI of 15, this must be mentioned in details as the reference did not mention it either.

It is not clear if the authors custom made their lipofundin, if so details must be written clearly about the amounts used of each ingredient listed.

line 97 "Imagination" of what?

The details of the confocal microscopy must be described and also detailed in the results section as nothing is clearly indicated in comparing of the images and the quantitative assays were not enough.

Results section needs revision as the figures are all not clear especially those of confocal microscopy. Figures must be revised, for instance Figure 1 is confusing, there is no need for two panels A and B, (also the panels are not labelled), it is better to make it all CFU/mL i.e. actual numbers instead of fold changes.

Abbreviations all over the manuscript should be revised as for instance FCS was mentioned as abbreviation line 63 before it was spelled out in line 67, the reverse should be done.

Based on all above, the conclusions were not supported by the poorly described results, the manuscript requires extensive re-writing and revision of its figures.

Author Response

Chen et al. manuscript discusses the potential effects of the use of lipid emulsions in clinical practice for IV injection in patients requiring IV nutrition through in vitro study on macrophage cell line (RAW264.7) infected with Staphylococcus aureus.

Starting from the abstract, it was difficult to read this manuscript especially the results that were unclear mainly due to the language and style. Thus, it was difficult to evaluate the manuscript with the use of such obscure language. Extensive editing of English language and style is required.

Ans: We have performed English editing through MDPI (english-36565).

The introduction does not provide sufficient background and needs major revision to its content as well as language.

Ans: We have modified our introduction and performed English editing through MDPI (english-36565).

The methods must be improved and details must be described for appropriate duplication of methods and hence results.

Ans: We have improved the details of Methods.

There was no mention of how the authors came up with a MOI of 15, this must be mentioned in details as the reference did not mention it either.

Ans: We have added the description about setting up a MOI of 15 in Materials and Methods (2.1.). Second paragraph, page 2: a 10-fold dilution of S. aureus suspension (OD600 = 1) was incubated with RAW264.7 cells for 30 min.

It is not clear if the authors custom made their lipofundin, if so details must be written clearly about the amounts used of each ingredient listed.

Ans: Thank you for the suggestion. We have added the details in Materials and Methods (2.2.). Third paragraph, page 2: Lipofundin 20% is composed of soybean oil (100 g/L), medium-chain triglycerides (MCT, 100 g/L), egg lecithin, glycerol, α-tocopherol, and sodium oleate. Lipofundin 20% was purchased from B. Braun Melsungen AG (Melsungen, Germany). ClinOleic 20% was obtained from Baxter (Norfolk, UK), which contains a mix of refined olive oil and refined soybean oil (200 g/L), glycerol, egg phospholipids, sodium oleate and sodium hydroxide. Omegaven (Fresenius-Kabi Austria GmbH, Graz, Austria) is a pure fish oil emulsion supplement (100 g/L) containing a high percentage of eicosapentaenoic acid (EPA) and docosahexaenoic (DHA). Di-α-tocopherol, glycerol, egg phosphatides, sodium oleate and sodium hydroxide are also included in it. The amount of Lipofundin (60 mg/ml) was chosen because the 10-fold concentration was applied to dissolve the clinical relevant concentration of anesthetics (propofol: 6 mg/ml).

line 97 "Imagination" of what?

Ans: We have corrected the mistake.

The details of the confocal microscopy must be described and also detailed in the results section as nothing is clearly indicated in comparing of the images and the quantitative assays were not enough.

Ans: We have mentioned more details about the confocal microscopy in Materials and Methods (2.6.) in page 3-4 and Results (3.3.) in page 7-9.

Results section needs revision as the figures are all not clear especially those of confocal microscopy. Figures must be revised, for instance Figure 1 is confusing, there is no need for two panels A and B, (also the panels are not labelled), it is better to make it all CFU/mL i.e. actual numbers instead of fold changes.

Ans: Thank you for the reminder. We have improved the quality of Figures. Figure 1 shown as fold changes is easily to compare the difference of bacterial survival between different groups.

Abbreviations all over the manuscript should be revised as for instance FCS was mentioned as abbreviation line 63 before it was spelled out in line 67, the reverse should be done.

Ans: Thank you for the reminder. We have checked and revised the abbreviations.

Based on all above, the conclusions were not supported by the poorly described results, the manuscript requires extensive re-writing and revision of its figures.

Ans: We have improved the quality of Figures and the descriptions of results to support our conclusions.

Round 2

Reviewer 2 Report

The authors failed to improve the results and the conclusions sections of the manuscript.

The data presented can not support the claim of the potential effects of the use of lipid emulsions in clinical practice as it was generated by an  in vitro study on mouse macrophage cell line (RAW264.7).

Thus it has to be very clearly indicated in the title, abstract, conclusions of the abstract, all figures especially figure 6 and in the conclusions that mice cell line macrophage were used and there are many limitations of their use, hence extrapolation to humans is not feasible with these experiments.

The authors failed to change figure 1 and 5 to be one panel figure with actual cfu data at 0 and 3 hours as requested previously in the first round of review. Providing derived fold hanges hides the actual data and statistical derivatives are misleading. Therefore, authors must let the reader see the actual bacterial cfu/ mL numbers to judge.

Figure 6 should be improved and mention that all was done in vitro

Abbreviations still need revision as the first mentions are abbreviated for instance ROS in  the abstract and introduction.

Discussion  needs improvements 

Author Response

The authors failed to improve the results and the conclusions sections of the manuscript.

The data presented can not support the claim of the potential effects of the use of lipid emulsions in clinical practice as it was generated by an  in vitro study on mouse macrophage cell line (RAW264.7).

Thus it has to be very clearly indicated in the title, abstract, conclusions of the abstract, all figures especially figure 6 and in the conclusions that mice cell line macrophage were used and there are many limitations of their use, hence extrapolation to humans is not feasible with these experiments.

Ans: According to the reviewer’ suggestion, we have modified the title, abstract, conclusions of the abstract, all figures especially figure 6 and the conclusions.

The authors failed to change figure 1 and 5 to be one panel figure with actual cfu data at 0 and 3 hours as requested previously in the first round of review. Providing derived fold hanges hides the actual data and statistical derivatives are misleading. Therefore, authors must let the reader see the actual bacterial cfu/ mL numbers to judge.

Ans: We have presented figure 1 and 5 with the actual bacterial cfu/ mL numbers.

Figure 6 should be improved and mention that all was done in vitro

Ans: We have modified the title of figure 6 as the following sentence.

“The in vitro inhibition model of lipid emulsions on S. aureus-infected mouse RAW264.7 macrophages”

Abbreviations still need revision as the first mentions are abbreviated for instance ROS in the abstract and introduction.

Ans: Thank you for your reminder. We have revised the abbreviation of ROS.

Discussion needs improvements 

Ans: We have improved the Discussion overall.